# Combating Client Dropout in Federated Learning via Friend Model Substitution

## Abstract

Federated learning (FL) is a new distributed machine learning framework known for its benefits on data privacy and communication efficiency. Since full client participation in many cases is infeasible due to constrained resources, partial participation FL algorithms have been investigated that *proactively* select/sample a subset of clients, aiming to achieve learning performance close to the full participation case. This paper studies a *passive* partial client participation scenario that is much less well understood, where partial participation is a result of external events, namely client dropout, rather than a decision of the FL algorithm. We cast FL with client dropout as a special case of a larger class of FL problems where clients can submit substitute (possibly inaccurate) local model updates. Based on our convergence analysis, we develop a new algorithm FL-FDMS that discovers friends of clients (i.e., clients whose data distributions are similar) on-the-fly and uses friends' local updates as substitutes for the dropout clients, thereby reducing the substitution error and improving the convergence performance. A complexity reduction mechanism is also incorporated into FL-FDMS, making it both theoretically sound and practically useful. Experiments on MNIST and CIFAR-10 confirmed the superior performance of FL-FDMS in handling client dropout in FL.

## 1 Introduction

Federated learning (FL) is a distributed machine learning paradigm where a set of clients with decentralized data work collaboratively to learn a model under the coordination of a centralized server. Depending on whether or not all clients participate in every learning round, FL is classified as either *full participation* or *partial participation*. While full participation is the ideal FL mode that achieves the best convergence performance, a lot of effort has been devoted to developing partial participation strategies via client selection/sampling Karimireddy et al. (2019); Li et al. (2019a); Yang et al. (2020); Ribero & Vikalo (2020); Chen et al. (2020a); Cho et al. (2020); Lai et al. (2021); Wu & Wang (2022); Balakrishnan et al. (2021) due to the attractive benefit of reduced resource (i.e. communication and computation) consumption. Existing works show that some of these partial participation strategies Li et al. (2019a); Yang et al. (2020) can indeed achieve performance close to full participation. Although the details differ, the principal idea of these strategies is the careful selection of *appropriate* clients to participate in each FL round. For example, in many cases Karimireddy et al. (2019); Li et al. (2019a); Yang et al. (2020), clients are sampled uniformly at random so that the participating clients form an "unbiased" representation of the whole client population in terms of the data distribution. In others Ribero & Vikalo (2020); Chen et al. (2020a); Cho et al. (2020); Lai et al. (2021); Wu & Wang (2022); Balakrishnan et al. (2021), "important" clients are selected more often to lead FL towards the correct loss descending direction.

This paper studies partial participation FL, but from an angle in stark contrast with existing works. In our considered problem, partial participation is a result of an arbitrary client dropout process, which the FL algorithm has absolutely no control over. For example, a client may not be able to participate (in other words, drop out) in a FL round due to, e.g., dead/low battery or loss of the communication signal. This means that the subset of clients participating in a FL round may not be "representative" or "important" in any sense. Note that the client dropout process by no means must be stochastic. Does FL still work

with arbitrary client dropout? How does client dropout affect the FL performance? How to mitigate the negative impact of client dropout on FL convergence? These are the central questions that this paper strives to answer.

We shall note that client dropout can occur simultaneously with client selection/sampling and hence partial participation can be a mixed result of both. As will become clear, our algorithm can be readily applied to this scenario and our theoretical results can also be extended provided that the client selection/sampling strategy used in conjunction has its own theoretical performance guarantee. However, since these results will depend on the specific client selection/sampling strategy adopted, and in order to better elucidate our main idea, this paper will not consider client selection/sampling.

**Main Contributions**. (1) Our study starts with a general convergence analysis for a class of FL problems where clients can submit substitute (inaccurate) local model updates for their true local model updates in each round. It turns out that this class of FL problems includes FL with client dropout as a special case. The main insight derived from this analysis is that the FL convergence performance depends on the total substitution error (i.e., the difference between the true update and its substitute) over the entire course of learning. As such, reducing the substitution error is the key to improving the FL performance with client dropout. (2) We then introduce the notion of "friendship" among clients, where clients are friends if their data distributions and hence their local model updates are similar enough. Therefore, an intuitive idea for mitigating the impact of client dropout is to use the local model update of a friend client (which does not drop out) as a substitute of that of the dropout client when computing the next global model, since doing so incurs a small substitution error. (3) Although the idea sounds promising, realizing it is highly non-trivial because the friendship is *unknown*. Thus, our algorithm must discover the friendship over the FL rounds and use the discovered friends for the local model update substitution purpose. We prove that our algorithm asymptotically achieves the same FL convergence bound when assuming that the friendship is fully revealed. Furthermore, because the vanilla friends discovery mechanism can involve a large number of model update comparison computations, we develop an improved algorithm that significantly reduces the computational complexity. With this improvement, our algorithm is both theoretically sound and practically useful. (4) Experiments on MNIST and CIFAR-10 confirmed that our algorithm can significantly improve the FL performance in the presence of client dropout.

## 2 Related Work

The FedAvg algorithm is an early FL algorithm proposed in konevcny et al. (2016), which sparked many follow-up works on FL. Early works focus on FL under the assumptions of i.i.d. datasets and full client participation Stich (2018); Yu et al. (2019b); Wang & Joshi (2018), and most of the theoretical works show a linear speedup for convergence for a sufficiently large number of learning rounds. For non-i.i.d. datasets, the performance of FedAvg and its variants were demonstrated empirically Karimireddy et al. (2019); Jeong et al. (2018); Zhao et al. (2018); Li et al. (2020b); Sattler et al. (2019), and convergence bounds were proven for the full participation case in Karimireddy et al. (2019); Stich et al. (2018); Yu et al. (2019a); Reddi et al. (2020). With partial client participation, the convergence of FedAvg was proven in Li et al. (2019a) for strongly convex functions, and the convergence of a generalized FedAvg with two-sided learning rates was analyzed in Karimireddy et al. (2019); Yang et al. (2020).

Most existing works on partial participation study strategies that *proactively* select or sample a subset of clients to participate in FL. The convergence of simple strategies such as random selection has been studied in Karimireddy et al. (2019); Li et al. (2019a); Yang et al. (2020). This analysis is relatively easy because of the "unbiased" nature of random selection. As a more sophisticated strategy, Cho et al. (2020) selects clients with higher local loss and proves an increased convergence speed. However, the vast majority of client selection strategies Ribero & Vikalo (2020); Lai et al. (2021); Wu & Wang (2022) only empirically shows the performance improvement. Very few works studied biased client selection in FL. In Cho et al. (2022), the authors presented a convergence rate bound under biased client selection, which contains a constant term due to the bias. In our paper, partial participation is *not* a decision of the FL algorithm, and our focus is on how to mitigate the negative impact of client dropout.

Client dropout is related to the "straggler" issue in FL, which is caused by the delayed local model uploading by some clients. Existing solutions to the straggler issue can be categorized into the following three types: doing nothing but waiting McMahan et al. (2017), allowing clients to upload their local models asynchronously to the server Wu et al. (2020); Li et al. (2019b); Xie et al. (2019); Chen et al. (2020b), and using the stored last updates of the inactive clients to join the model aggregation Yan et al. (2020); Gu et al. (2021). A straggler-resilient FL is proposed in Reisizadeh et al. (2020) that incorporates statistical characteristics of the clients' data to adaptively select clients. Some other recent works Ruan et al. (2021); Yang et al. (2022) studied FL with flexible client participation where client dropout is part of the considered scenario and provided general convergence analysis. However, unlike our paper, these works Ruan et al. (2021); Yang et al. (2022) do not propose mitigation schemes for client dropout.

Our proposed algorithm exploits client similarity for model substitution, which is related to clustered FL Ghosh et al. (2020); Ruan & Joe-Wong (2022); Dennis et al. (2021) which clusters clients with similar data distributions into the same group. However, our algorithm has a very different motivation and objective. Moreover, our algorithm only requires computing pair-wise similarity, which can be computed easily on the server, whereas clustered FL needs to identify a set of clients belonging to the same cluster, which incurs much higher computation and communication overhead.

## 3   Federated Learning with Client Dropout

We consider a server and a set of $K$ clients, indexed by $\mathcal{K} = \{1, ..., K\}$, who work together to train a machine learning model by solving a distributed optimization problem:

$$\min_{w \in \mathbb{R}^d} \left\{ f(w) := \frac{1}{K} \sum_{k=1}^{K} \mathbb{E}_{\xi^k \sim \mathcal{D}^k}[F^k(w; \xi^k)] \right\} \tag{1}$$

where $F^k : \mathbb{R}^d \to \mathbb{R}$ denotes the objective function, $\xi^k \sim \mathcal{D}^k$ represents the sample/s drawn from distribution $\mathcal{D}^k$ at the $k$-th client and $w \in \mathbb{R}^d$ is the model parameter to learn. In a non-i.i.d. data setting, which is the focus of this paper, the distributions $\mathcal{D}^k$ are different across the clients.

We consider a typical FL algorithm konevcny et al. (2016) working in the client dropout setting. In each round $t$, only a subset $\mathcal{S}_t \subseteq \mathcal{K}$ of clients participate due to external reasons uncontrollable by the FL algorithm. We call the clients that cannot participate or complete the task *dropout* (or *inactive*) clients. Then, FL executes the following four steps among the *non-dropout* (or *active*) clients in round $t$:

1. **Global model download**. Each client $k \in \mathcal{S}_t$ downloads the global model $w_t$ from the server.

2. **Local model update**. Each client $k \in \mathcal{S}_t$ uses $w_t$ as the initial model to train a new local model $w_{t+1}^k$, typically by using mini-batch stochastic gradient descent (SGD) as follows:

$$\begin{aligned} w_{t,0}^k &= w_t \\ w_{t,\tau+1}^k &= w_{t,\tau}^k - \eta_L g_{t,\tau}^k, \forall \tau = 1, ..., E \\ w_{t+1}^k &= w_{t,E}^k \end{aligned} \tag{2}$$

   where $\xi_{t,\tau}^k$ is a mini-batch of data samples independently sampled uniformly at random from the local dataset of client $k$, $g_{t,\tau}^k = \nabla F^k(w_{t,\tau}^k; \xi_{t,\tau}^k)$ is the mini-batch stochastic gradient, $\eta_L$ is the client local learning rate and $E$ is the number of epochs for local training.

3. **Local model upload**. Clients upload their local model updates to the server. Instead of uploading the local model $w_{t+1}^k$ itself, client $k$ can simply upload the *local model update* $\Delta_t^k$, which is defined as the accumulative model parameter difference as follows:

$$\Delta_t^k = -\sum_{\tau=0}^{E-1} g_{t,\tau}^k \tag{3}$$

4. **Global model update**. The server updates the global model by using the aggregated local model updates of the clients in $\mathcal{S}_t$:

$$w_{t+1} = w_t + \eta\eta_L\Delta_t, \quad \text{where} \quad \Delta_t := \frac{1}{S_t} \sum_{k \in \mathcal{S}_t} \Delta_t^k \tag{4}$$

and $\eta$ is the global learning rate and $S_t \triangleq |\mathcal{S}_t|$ denotes the number of the non-dropout clients.

For the main result of this paper, we consider the most general case of the client dropout process by imposing only an upper limit on the dropout ratio. That is, there exists a constant $\alpha \in [0,1)$ such that $(K - S_t)/K \leq \alpha$. If all clients drop out in a round, then essentially the round is skipped. Later in this paper, we will impose additional conditions on the dropout process to facilitate our theoretical analysis.

Also note that if $\mathcal{S}_t$ were a choice of the FL algorithm, then the problem would become FL with client selection/sampling. We stress again that in our problem, $\mathcal{S}_t$ is not a choice, it is an uncontrollable client participation scenario.

## 4 Convergence Analysis

Consider an FL round $t$ where the set $\mathcal{S}_t$ of clients are active while the remaining set $\mathcal{K}\backslash\mathcal{S}_t$ of clients dropped out. Thus, one can only use the local model updates $\Delta_t^k$ of the active clients in $\mathcal{S}_t$ to perform global model updates since the inactive clients upload nothing to the server. However, rather than completely ignoring the inactive clients, we write the aggregate model update $\Delta_t$ in a different way to include all clients in the equation:

$$\Delta_t := \frac{1}{S_t} \sum_{k \in \mathcal{S}_t} \Delta_t^k = \frac{1}{K} \left( \sum_{k \in \mathcal{S}_t} \Delta_t^k + \sum_{k \in \mathcal{K}\backslash\mathcal{S}_t} \tilde{\Delta}_t^k \right) \tag{5}$$

where in the second equality we simply take $\tilde{\Delta}_t^k = \frac{1}{S_t}\sum_{k\in\mathcal{S}_t}\Delta_t^k$. In other words, although the inactive clients did not participate in the round $t$'s learning, it is equivalent to the case where an inactive client $k \in \mathcal{K}\backslash\mathcal{S}_t$ uses $\tilde{\Delta}_t^k = \frac{1}{S_t}\sum_{k\in\mathcal{S}_t}\Delta_t^k$ as a substitute of its true local update $\Delta_t^k$ (which it may not even calculate due to dropout). Apparently, because $\tilde{\Delta}_t^k \neq \Delta_t^k$ in general, similar substitutes lead to a biased error in the global update and hence affect the FL convergence performance.

Leveraging the above observation, we consider a larger class of FL problems that include client dropout as a special case. Specifically, imagine that an inactive client $k$, instead of contributing nothing, uses a substitute $\tilde{\Delta}_t^k$ for $\Delta_t^k$ when submitting its local model update. Apparently, $\tilde{\Delta}_t^k = \frac{1}{S_t}\sum_{k\in\mathcal{S}_t}\Delta_t^k$ is a specific choice of the substitute. We will still use the notation $\Delta_t$ as the aggregate model update with local update substitution and the readers should not be confused. Our convergence analysis will utilize the following standard assumptions about the FL problem.

**Assumption 1 (Lipschitz Smoothness)** *The local objective functions satisfy the Lipschitz smoothness property, i.e.,$\exists L > 0$, such that $\|\nabla F^k(x) - \nabla F^k(y)\| \leq L\|x - y\|$, $\forall x, y \in \mathbb{R}^d$ and $\forall k \in \mathcal{K}$.*

**Assumption 2 (Unbiased Local Gradient Estimator)** *The mini-batch based local gradient estimator is unbiased, i.e. $\mathbb{E}_{\xi^k \sim \mathcal{D}^k}[\nabla F^k(x; \xi^k)] = \nabla F^k(x)$, $\forall k \in \mathcal{K}$.*

**Assumption 3 (Bounded Local and Global Variance)** *There exist constants $\rho_L > 0$ and $\rho_G > 0$ such that the variance of each local gradient estimator is bounded, i.e., $\mathbb{E}_{\xi^k \sim \mathcal{D}^k}\left[\|\nabla F^k(x; \xi^k) - \nabla F^k(x)\|^2\right] \leq \rho_L^2, \forall x, \forall k \in \mathcal{K}$. And the global variability of the local gradient is bounded by $\|\nabla F^k(x) - \nabla f(x)\|^2 \leq \rho_G^2, \forall x, \forall k \in \mathcal{K}$.*

The following result on the upper bound for the $\tau$-step SGD under full participation will be used.

**Lemma 1 (Lemma 4 in Reddi et al. (2020))** *For any step-size satisfying $\eta_L \leq \frac{1}{8LE}$, we have: $\forall \tau = 0, ..., E-1$*

$$\frac{1}{K}\sum_{k=1}^{K}\mathbb{E}[\|w_{t,\tau}^k - w_t\|^2] \leq 5E\eta_L^2(\rho_L^2 + 6E\rho_G^2) + 30E^2\eta_L^2\|\nabla f(w_t)\|^2 \tag{6}$$

Several additional notations will be handy. Let $\bar{\Delta}_t := \frac{1}{K}\sum_{k\in\mathcal{K}}\Delta_t^k$ be the average local model update assuming that all clients are active in round $t$ and submitted their true local updates. Thus, $e_t := \Delta_t - \bar{\Delta}_t$ represents aggregate global update error due to client dropout and local update substitution in round $t$. Furthermore, let $e_t^k := \tilde{\Delta}_t^k - \Delta_t^k$, $\forall k \in \mathcal{K}\backslash\mathcal{S}_t$ be the individual substitution error for an individual inactive client $k$ in round $t$.

**Theorem 1** *Let constant local and global learning rates $\eta_L$ and $\eta$ be chosen as such that $\eta_L \leq \frac{1}{8EL}$ and $\eta\eta_L \leq \frac{1}{4EL}$. Under Assumption 1-3, the sequence of model $w_t$ generated by using model update substitution with a substitution error sequence $e_0, ..., e_{T-1}$ satisfies*

$$\min_{t=0,...,T-1}\mathbb{E}\|\nabla f(w_t)\|^2 \leq \frac{f_0 - f_*}{c\eta\eta_L ET} + \Phi + \Psi(e_0, ..., e_{T-1}) \tag{7}$$

*where $\Phi = \frac{1}{c}\left[5\eta_L^2 EL^2(\rho_L^2 + 6E\rho_G^2) + \frac{\eta\eta_L L}{K}\rho_L^2\right]$, $c$ is a constant, $f_0 \triangleq f(w_0)$, $f_* \triangleq f(w_*)$, $w_*$ is the optimal model and*

$$\Psi(e_0, ..., e_{T-1}) = \frac{1 + 3\eta\eta_L LE}{cE^2 T}\sum_{t=0}^{T-1}\mathbb{E}_t[\|e_t\|^2] \tag{8}$$

*where the expectation is over the local dataset samples among the clients.*

The above convergence bound contains three parts: a vanishing term $\frac{f_0 - f_*}{c\eta\eta_L ET}$ as $T$ increases, a constant term $\Phi$ whose size depends on the problem instance parameters and is independent of $T$, and a third term that depends on the sequence of substitution errors $e_0, ..., e_{T-1}$. Thus, by using constant learning rates $\eta$ and $\eta_L$ and assuming that $\|e_t\|^2$ is bounded, then the convergence bound is $O(1/T) + C$, where $C$ is a constant. The key insight derived by Theorem 1 is that the FL convergence bound depends on the cumulative substitution error $\sum_{t=0}^{T-1}\mathbb{E}_t[\|e_t\|^2]$. When there is no dropout client, namely all clients participated in every round and submitted their true local model updates, the cumulative substitution error is 0 and hence, $\Psi(e_0, ..., e_{T-1}) = 0$. Thus, the convergence bound is simply $\frac{f_0 - f_*}{c\eta\eta_L ET} + \Phi$, which degenerates to the same bound established in Yang et al. (2020) for the normal full participation case.

We want to clarify that we did not intend to introduce a new convergence analysis technique for FL. Instead, our work's novelty lies in formalizing the effects of client dropout and, more broadly, model substitution on FL convergence and, based on this understanding, developing mitigation methods to enhance convergence. Our analysis provides an intuitive understanding of the impact of client dropout and model substitution on FL convergence and characterizes the convergence under a biased scenario, which is an important and previously unaddressed issue in FL research. The main challenge was consolidating all the bias-related errors into a single term in the final convergence bound.

Next, we derive a more specific bound on $\Psi(e_0, ..., e_{T-1})$ for the naive dropout case where an inactive client uploads nothing to the server or, equivalently, uses $\frac{1}{S_t}\sum_{k\in\mathcal{S}_t}\Delta_t^k$ as a substitute. The following additional assumption is needed.

**Assumption 4** *For any two clients $i$ and $j$, the local model update difference is bounded as follows:*

$$\mathbb{E}[\|\Delta^i(w) - \Delta^j(w)\|^2] \leq \sigma_{i,j}^2, \forall w \tag{9}$$

*where the expectation is over the local dataset samples.*

Assumption 4 provides a pairwise characterization of clients' dataset heterogeneity in terms of the local model updates. When two clients $i, j$ have the same data distribution and assuming that the min-batch SGD utilizes the entire local dataset (i.e., the local gradient estimator is accurate), then it is obvious $\sigma_{i,j}^2 = 0$. We let $\sigma_P^2 \triangleq \max_{i,j} \sigma_{i,j}^2$ be the maximum pairwise difference.

With Assumption 4, the round-$t$ substitution error can then be bounded as $\mathbb{E}[\|e_t\|^2] \leq \alpha^2 \sigma_P^2$ (See details in Appendix A.4). Plugging this bound into $\Psi(e_0, ..., e_{T-t})$, we have

$$\Psi(e_0, ..., e_{T-1}) \leq \frac{\alpha^2 \sigma_P^2 (1 + 3\eta\eta_L LE)}{cE^2} \triangleq \bar{\Psi} \tag{10}$$

Note that $\bar{\Psi}$ is a constant independent of $T$. This implies that, with constant learning rates $\eta_L$ and $\eta$, $\min_t \mathbb{E}\|\nabla f(w_t)\|^2$ converges to some value at most $\Phi + \bar{\Psi}$ as $T \to \infty$.

**Convergence Rate**: By using a local learning rate $\eta_L \sim \mathcal{O}(\frac{1}{\sqrt{T}})$ and a global learning rate $\eta \sim \mathcal{O}(1)$, the convergence rate can be improved to

$$\mathcal{O}(\frac{1}{E\sqrt{T}}) + \underbrace{\mathcal{O}(\frac{E^2}{T}) + \mathcal{O}(\frac{1}{K\sqrt{T}})}_{\Phi} + \underbrace{\mathcal{O}(\frac{\alpha^2 \sigma_P^2}{\sqrt{T}}) + \mathcal{O}(\frac{\alpha^2 \sigma_P^2}{E})}_{\bar{\Psi}} \tag{11}$$

Similar to the biased client selection in Cho et al. (2022), even with decaying learning rates, the convergence bound contains a non-vanishing term (i.e., the last term) in the general case. However, in our case, the non-vanishing term is due to replacing the model of the dropout clients with an arbitrary model. Nevertheless, with a decreasing and vanishing dropout rate $\alpha$, the last term also vanishes and the $\min_t \mathbb{E}\|\nabla f(w_t)\|^2$ converges to the stationary point.

# 5 Friend Model Substitution

In this section, we develop a new algorithm to reduce or even eliminate the non-vanishing term in the convergence bound of FL with client dropout. Our key idea is to find a better substitute $\tilde{\Delta}_t^k$ for $\Delta_t^k$ when client $k$ drops out in round $t$ in order to reduce $\Psi(e_0, ..., e_{T-1})$. This is possible by noticing that $\sigma_{i,j}^2$ are different across client pairs and the local model updates are more similar when the clients' data distributions are more similar. Thus, when a client $i$ drops out, one can use the local model update $\Delta_t^j$ as a replacement of $\Delta_t^i$ if $j$ shares a similar data distribution with $i$, or in our terminology, $j$ is a friend of $i$. We make "friendship" formal in the following definition.

**Definition 1 (Friendship)** *Let $\sigma_F^2 < \sigma_P^2$ be some constant. We say that clients $i$ and $j$ are friends if $\sigma_{i,j}^2 \leq \sigma_F^2$. Further, denote $\mathcal{B}_k$ as the set of friends of client $k$ and $B_k = |\mathcal{B}_k|$ as the size of $\mathcal{B}_k$.*

**Assumption 5 (Friend Presence)** *In any round $t$, for any inactive client $i \in \mathcal{K} \backslash \mathcal{S}_t$, there exists at least one active client $j$ that is client $i$'s friend.*

Assumption 5 states that using the local model update of a friend to replace that of an inactive client is feasible. Nevertheless, this assumption can be **relaxed** so that friends may not be present in every round when a client drops out. We will cover the **relaxed** case in Appendix B.

Note that although such "friendship" exists among the clients, they are **hidden** from the FL algorithm. Thus, it is not straightforward to substitute the model update of a dropout client with that of its friends. Nevertheless, it is still useful to understand what can be achieved assuming that the friendship information is fully revealed. The convergence bound in this *non-realizable* case will serve as an optimal baseline for our algorithm to be developed.

With friend model substitution, the accumulated substitution error can be bounded as $\mathbb{E}[\|e_t\|^2] \leq \alpha^2 \sigma_F^2$ (see Appendix A.4). Substituting this bound into $\Psi(e_0, ..., e_{T-1})$, we have

$$\Psi(e_0, ..., e_{T-1}) \leq \frac{\alpha^2 \sigma_F^2 (1 + 3\eta\eta_L LE)}{cE^2} \triangleq \Psi^* \tag{12}$$

Note that $\Psi^*$ is still a constant independent of $T$ but it is much smaller than $\bar{\Psi}$, since typically $\sigma_F^2 \ll \sigma_P^2$. Therefore, the convergence bound can be significantly improved if the algorithm utilizes the friendship information.

**Convergence Rate**: With a local learning rate $\eta_L \sim \mathcal{O}(\frac{1}{\sqrt{T}})$ and a global learning rate $\eta \sim \mathcal{O}(1)$, the convergence rate can be improved to

$$\mathcal{O}(\frac{1}{E\sqrt{T}}) + \underbrace{\mathcal{O}(\frac{E^2}{T}) + \mathcal{O}(\frac{1}{K\sqrt{T}})}_{\Phi} + \underbrace{\mathcal{O}(\frac{\alpha^2\sigma_F^2}{\sqrt{T}}) + \mathcal{O}(\frac{\alpha^2\sigma_F^2}{E})}_{\Psi^*} \tag{13}$$

The above convergence bound still has a non-vanishing constant term (i.e., the last term) in the general case. In some special cases where $\sigma_F^2$ dynamically decreases over round, this term may also vanish as $T \to \infty$. For example, consider that clients are grouped into clusters and clients from the same cluster have identical data distribution. Borrowing the idea from Shi et al. (2022) on the increasing mini-batch size, the local model update difference of two clients, i.e., $\sigma_{i,j}^2$ from the same cluster also decreases with an increasing batch size $B$. Assuming $\mathbb{E}[\|\Delta^i(w) - \Delta^j(w)\|^2] \leq \sigma_F^2/B, \forall w$, and by using the mini-batch size sequence $B_t = B_0 t$ where $B_0$ is the initial mini-batch size, then the last term becomes $\mathcal{O}(\frac{\alpha^2\sigma_F^2 \ln T}{TE})$, which goes to 0 as $T \to \infty$. Therefore, the constant term in the convergence rate is eliminated even with a constant dropout rate $\alpha$.

The above analysis shows that the FL convergence can be substantially improved if the algorithm can utilize the friendship information. Next, we develop a learning-assisted FL algorithm, called FL with Friend Discovery and Model Substitution (FL-FDMS), that discovers the friends of clients and uses the model update of the discovered friends for substitution. We prove that FL-FDMS achieves asymptotically the optimal error bound $\Psi^*$.

## 5.1 Algorithm (FL-FDMS)

In each FL round $t$, in addition to the regular steps in the FL algorithm described in Section 3, FL-FDMS performs different actions depending on whether or not a client drops out. For active clients, FL-FDMS calculates pairwise similarity scores to learn the similarity between clients. For inactive clients, FL-FDMS uses the historical similarity scores to find active friend clients and use their local model updates as substitutes. We describe these two cases in more detail below.

**Active Clients**. For any pair of active clients $i$ and $j$ in $\mathcal{S}_t$. The server calculates a similarity score $r_t^{i,j} = r(\Delta_t^i, \Delta_t^j)$ based on their uploaded local model updates $\Delta_t^i$ and $\Delta_t^j$. Many functions can be used to calculate the score. For example, $r(\Delta_t^i, \Delta_t^j)$ can simply be the negative model difference, i.e., $-\|\Delta_t^i - \Delta_t^j\|$, or the normalized cosine similarity, i.e.,

$$r(\Delta_t^i, \Delta_t^j) = \frac{1}{2}\left(\frac{\langle\Delta_t^i, \Delta_t^j\rangle}{\|\Delta_t^i\|\|\Delta_t^j\|} + 1\right) \tag{14}$$

Because the normalized cosine similarity takes value from a bounded and normalized range $[0, 1]$, which is more amenable for mathematical analysis, we will use this function in this paper. Clearly, a higher similarity score implies that the two clients are more similar in terms of their data distribution.

However, a single similarity score calculated in one particular round does not provide accurate similarity information because of the randomness in the initial model in that round and the randomness in the mini-batch SGD for local model computation. Thus, the server maintains and updates an average similarity score $R_t^{i,j}$ for clients $i$ and $j$ based on all similarity scores calculated so far as follows,

$$R_t^{i,j} = \begin{cases} \frac{N_{t-1}^{i,j}}{N_{t-1}^{i,j}+1}R_{t-1}^{i,j} + \frac{1}{N_{t-1}^{i,j}+1}r_t^{i,j}, & \text{if } i,j \in \mathcal{S}_t \\ R_{t-1}^{i,j}, & \text{otherwise} \end{cases} \tag{15}$$

where $N_t^{i,j}$ is the number of rounds where both clients $i$ and $j$ did not drop out up to round $t$.

**Inactive Clients**. For any inactive client $k$, the server looks up $R_t^{k,i}$ between $k$ and every active client $i \in \mathcal{S}_t$, finds the one with the highest similarity score, denoted by $\phi_t(k) = \arg\max_{i \in \mathcal{S}_t} R_t^{k,i}$, and uses the local model update $\Delta_t^{\phi_t(k)}$ as a substitute for $\Delta_t^k$ when computing the global update.

**Remark on Privacy**: The averaged similarity score is calculated based on the uploaded model and does not require any additional information from the client. Previous works such as Ruan & Joe-Wong (2022); Ghosh et al. (2020), albeit addressing different FL problems, also rely on finding the client relationships or clusters. Thus, the privacy protection level of our algorithm is similar to that of those algorithms.

## 5.2 Convergence Analysis

In this subsection, we analyze the convergence of FL-FDMS. Since Theorem 1 has already proven the FL convergence bound depending on a general error sequence, we will focus only on bounding $\Psi(e_0, e_1, ..., e_{T-1})$ in our algorithm. The following additional assumptions on the dropout process are needed.

**Assumption 6 (Sufficiently Many Common Rounds)** *There exists a constant $\beta \in (0, 1]$ so that for any pair of clients $i$ and $j$, $N_t^{i,j}$ satisfies $N_t^{i,j} \geq \beta t, \forall t$.*

We first establish a bound on the probability that $\phi_t(k)$ selected by our algorithm is not a friend of client $k$. Let $\mathbb{E}_t[r_t^{i,j}] = \mu^{i,j}$ be the expected similarity score between clients $i$ and $j$. Thus $\delta_k = \min_{i \in \mathcal{B}_k} \mu^{k,i} - \max_{i \notin \mathcal{B}_k} \mu^{k,i}$ denotes the minimum similarity score gap between a friend client and a non-friend client. We further let $\delta_{\min} = \min_k \delta_k$ for all $k \in \mathcal{K}$.

**Lemma 2** *The probability that our algorithm selects a non-friend client for an inactive client in round $t$ is upper bounded by $2K \exp\left(\frac{-\beta \delta_{min}^2 t}{2}\right)$.*

Now, we are ready to bound $\mathbb{E}\|e_t\|^2$ by using FL-FDMS. Firstly, we can show the following bound on $\mathbb{E}\|e_t\|^2$ (see Appendix A.4):

$$\mathbb{E}\|e_t\|^2 \leq \alpha^2 \left(\sigma_F^2 + 2K \exp\left(\frac{-\beta \delta_{min}^2 t}{2}\right)(\sigma_P^2 - \sigma_F^2)\right) \tag{16}$$

Plugging this bound into $\Psi$ (see Appendix A.5), we have

$$\Psi(e_0, ..., e_{T-1}) \leq \Psi^* + 2K\bar{\Psi} \frac{1 - \exp\left(\frac{-\beta \delta_{\min}^2 T}{2}\right)}{T\left(1 - \exp\left(\frac{-\beta \delta_{\min}^2}{2}\right)\right)} \tag{17}$$

Note that as $T \to \infty$, $\Psi(e_0, ..., e_{T-1}) \to \Psi^*$. Therefore, FL-FDMS asymptotically approaches the convergence bound obtained in the fully revealed friendship information case.

## 5.3 Reducing Similarity Computation Complexity

FL-FDMS requires pairwise similarity score computation for all active clients in every round, causing a big computation burden when $S_t$ is large. We now design a simple mechanism inspired by Hoeffding races Maron & Moore (1993) to reduce the computational complexity and incorporate it in FL-FDMS. To this end, we introduce a notion called candidate (friend) set $\mathcal{C}_t^k$ for each client $k$, which keeps the potential friend list of client $k$. Initially, $\mathcal{C}_t^k$ is the entire client set $\mathcal{K}$, but over time, $\mathcal{C}_t^k$ shrinks by eliminating non-friend clients with high probability. Thus, the server only needs to compute the similarity score of clients in the candidate friend set of a client $k$ and pick a client from this set for substitution purposes. The updating rule for $\mathcal{C}_t^k$ is as follows. In every round $t$, for each client $k$: (1) Find the client $\phi_t(k)$ with the highest similarity score, i.e., $\phi_t(k) = \arg\max_{i \in \mathcal{C}_t^k} R_t^{k,i}$. (2) Compute the similarity score gap $g_{k,i} = R_t^{k,\phi_t(k)} - R_t^{k,i}$ for any other client $i \neq \phi_t(k), i \in \mathcal{C}_t^k$. (3) Eliminate $i$ from $\mathcal{C}_t^k$ if $g_{k,j} \geq \Theta_t$, namely $\mathcal{C}_{t+1}^k \leftarrow \mathcal{C}_t^k - \{i : g_{k,i} \geq \Theta_t\}$, where $\Theta_t$ is a threshold parameter decreasing over $t$. In Theorem 2, we design a specific threshold sequence $\Theta_t$ and prove that our convergence bound in the previous section holds with high probability.

**Theorem 2** *Let $\delta_f = \max_k\{\max_{i\in\mathcal{B}_k}\mu^{k,i} - \min_{i\in\mathcal{B}_k}\mu^{k,i}\}$ be the maximum similarity score gap among friends of any client, and $B_{\max} = \max_k B_k$ be the maximum number of friends that any client can have. For any $p \in (0,1)$, by setting $\Theta_t = \sqrt{\frac{2\ln(2K^2TB_{\max})-2\ln p}{\beta t}} + \delta_f$, FL-FDMS with complexity reduction yields, with probability at least $1-p$, the same bound on $\Psi$ as in equation 74.*

Note that in order to establish the convergence bound, we used several loose bounds in the proof of Theorem 2, thereby resulting in an unnecessarily large threshold sequence $\Theta_t$. In practice, the threshold sequence $\Theta_t$ can be chosen much smaller than what is given in Theorem 2, and hence non-friend clients can be eliminated more quickly with high confidence.

## 6 Experiments

### 6.1 Setup

We use Python3 and the Pytorch library, and our code is adapted from Jadhav (2020), which is under the MIT License. The experiments were run on an Ubuntu 18.04 machine with an Intel Core i7-10700KF 3.8GHz CPU and GeForce RTX 3070 GPU. All experiment results are averaged over 10 repeats.

We perform experiments on two standard public datasets, namely MNIST and CIFAR-10, which are widely used in FL experiments, in a clustered setting as well as a general setting. In the clustered settings (one on MNIST and one on CIFAR-10), we artificially create 5 client clusters where clients in the same cluster possess data samples with the same labels. Thus, clients in the same cluster are naturally regarded as friends. However, the clustering structure is *unknown* to our algorithm. Such a clustering setting provides a controlled environment for us to evaluate the friend discovery performance of FL-FDMS. In the general setting (on CIFAR-10), 20 clients receive a random subset of the whole dataset using a common way of generating non-iid FL datesets that is widely used in existing works.

#### 6.1.1 FL Dataset

**Clustered Setting - MNIST**: The MNIST dataset has 60000 training data samples with 10 classes. The training dataset is first split into 10 sub-datasets with samples in the same sub-dataset having the same label. There are 20 clients which are grouped into 5 client clusters with an equal number of clients. Each client cluster is associated with 2 randomly drawn sub-datasets. Then each client randomly draws 200 samples from its corresponding two sub-datasets. This approach to creating the FL dataset was introduced in a recent clustered FL work Ghosh et al. (2020).

**Clustered Setting - CIFAR-10**: The CIFAR-10 dataset has 50000 training data samples with 10 classes. The training dataset is first split into 10 sub-datasets with samples in the same sub-dataset having the same label. There are 20 clients which are grouped into 5 client clusters with an equal number of clients. Each client cluster is associated with 2 randomly drawn sub-datasets. Then each client randomly draws 1000 samples from its corresponding two sub-datasets.

**General Setting - CIFAR-10**: The CIFAR-10 dataset has 50000 training data samples. After shuffling the samples in label order, all samples are divided into 200 partitions with each partition having 250 samples. There are 20 clients. Each client then randomly picks 2 partitions. This method is a common way of generating non-i.i.d. FL dataset, which is widely used in the existing works McMahan et al. (2017); Li et al. (2021)

#### 6.1.2 FL Models

**MNIST**: The CNN model has two $5 \times 5$ convolution layers, a fully connected layer with 320 units and ReLU activation, and a final output layer with softmax. The first convolution layer has 10 channels while the second one has 20 channels. Both layers are followed by $2 \times 2$ max pooling. The following parameters are used for training: the local batch size $BS = 5$, the number of local iterations $E = 2$, the local learning rate $\eta_L = 0.1$ and the global learning rate $\eta = 0.1$.

**CIFAR-10**: The CNN model has two $5 \times 5$ convolution layers, three fully connected layers and ReLU activation, and a final output layer with softmax. The following parameters are used for training: the local batch size $BS = 20$, the number of local iterations $E = 2$, the local learning rate $\eta_L = 0.1$ and the global learning rate $\eta = 0.1$.

### 6.1.3 FL Benchmarks

We compare FL-FDMS with three variants of FedAvg since in this paper we describe FL-FDMS in the context of FedAvg.

**Full Participation (Full)**. This is the ideal case where all clients participate in FL without dropout. It is used as a performance upper bound.

**Client Dropout (Dropout)**. In this case, the server simply ignores the dropout clients and performs global aggregation on the non-dropout clients.

**Staled Substitute (Stale)**. Another method to deal with dropout clients is to use their last uploaded local model updates for the current round's global aggregation. Such a method was also used to deal with the "straggler" issue in FL in some previous works Yan et al. (2020); Gu et al. (2021). Two apparent drawbacks of this method are, firstly, the local model updates can be very outdated if a client keeps dropping out, and secondly, the server has to keep a copy of the most recent local model update for every client, thereby incurring a large storage cost when the number of clients is large.

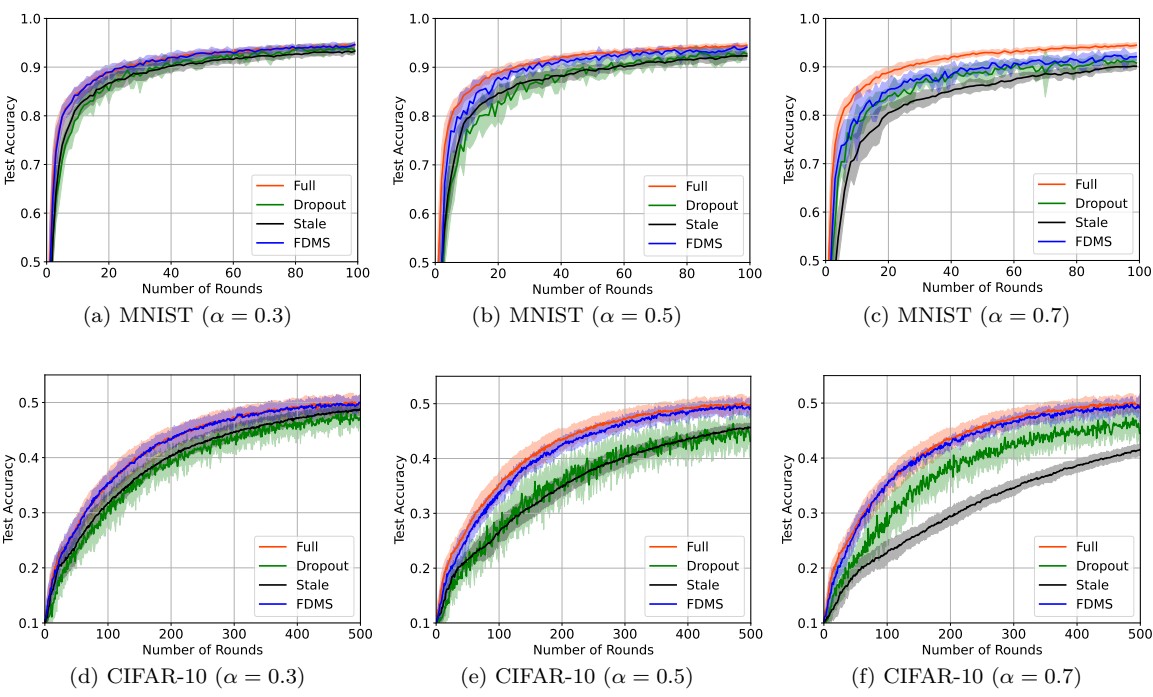

Figure 1: Performance comparison on the clustered setting with various $\alpha$

## 6.2 Performance Comparison

We first compare the convergence performance in the clustered setting under different dropout ratios $\alpha \in \{0.3, 0.5, 0.7\}$. Fig. 1 plots the convergence curves on the MNIST dataset and the CIFAR-10 dataset, respectively. Several observations are made as follows. First, **FL-FDMS** outperforms **Dropout** and **Stale** in terms of test accuracy and convergence speed and achieves performance close to **Full** in all cases. Second, **FL-FDMS** reduces the fluctuations caused by the client dropout on the convergence curve. Third, with

a larger dropout ratio, the performance improvement of **FL-FDMS** is larger. Fourth, on more complex datasets (e.g., CIFAR-10), **FL-FDMS** achieves an even more significant performance improvement.

We note that **Stale** is the most sensitive to the dropout ratio $\alpha$ and its performance degrades significantly as $\alpha$ increases. This is because with a larger $\alpha$, individual clients have few participating opportunities. As a result, the staled local models become too outdated to provide useful information for the current round's global model updating.

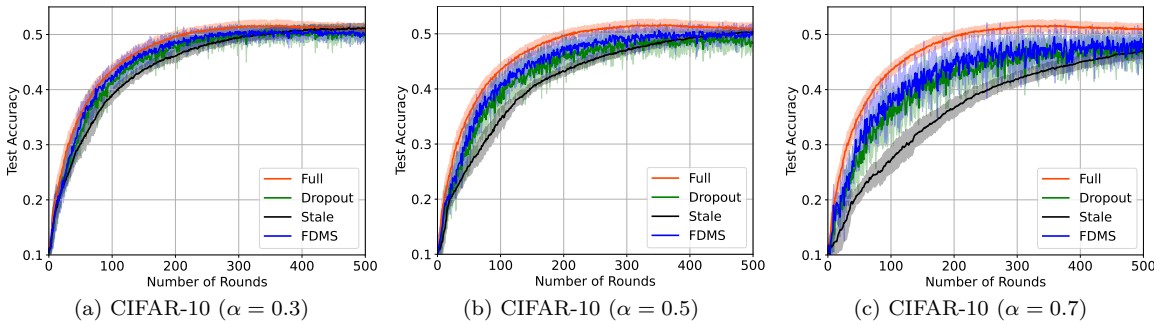

| (a) CIFAR-10 ($\alpha = 0.3$) | (b) CIFAR-10 ($\alpha = 0.5$) | (c) CIFAR-10 ($\alpha = 0.7$) |

Figure 2: Performance comparison on the CIFAR-10 general setting with various $\alpha$

We also perform experiments in the more general non-iid case to illustrate the wide applicability of the proposed algorithm. Fig. 2 plots the convergence curves on CIFAR-10 under the general setting. The results confirm the superiority of **FL-FDMS**. However, we also note that the improvement is smaller than that in the clustered setting. This suggests a limitation of **FL-FDMS**, which works best when the "friendship" relationship among the clients is stronger.

### 6.3  Friend Discovery

**FL-FDMS** relies on successfully discovering the friends of dropout clients. In Fig. 3, we show the pairwise similarity scores in the final learning round. In our controlled clustered setting, 20 clients were grouped into 5 clusters, but this information was not known by the algorithm at the beginning. As the figure shows, the similarity scores obtained by **FL-FDMS** are larger for intra-cluster client pairs and smaller for inter-cluster client pairs, indicating that the clustering/friendship information can be successfully discovered. Moreover, our experiments show that the discovered friendship is more obvious for CIFAR-10 than for MNIST. This is likely due to the different dataset structures and the different CNN models adopted.

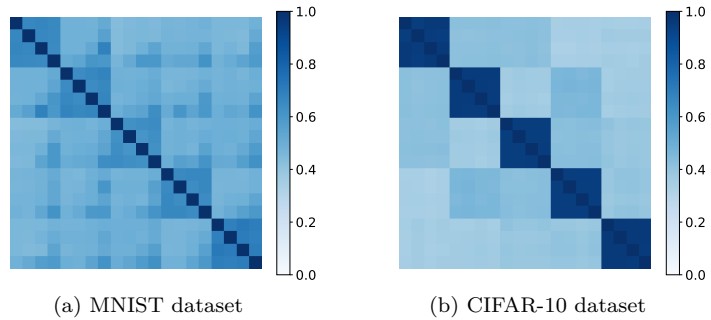

| (a) MNIST dataset | (b) CIFAR-10 dataset |

Figure 3: Pairwise similarity scores in the final learning round.

### 6.4  Computation Complexity

In this set of experiments, we investigate the computational complexity of **FL-FDMS** with and without the complexity reduction mechanism (CR) on CIFAR-10 in the clustered setting. For the complexity reduction

mechanism, we use two different threshold sequences: $\Theta_t$ in Theorem 2 (Case 1), and $0.5\Theta_t$ (Case 2). In Fig. 4, we can see that the complexity reduction mechanism can significantly reduce the computation complexity while still achieving a similar learning performance. We also notice that a smaller threshold sequence accelerates the client elimination process while keeping a similar learning performance.

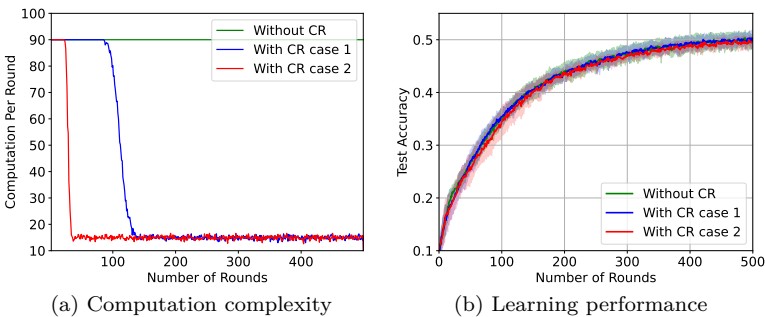

(a) Computation complexity    (b) Learning performance

Figure 4: Complexity reduction on the CIFAR-10 clustered setting ($\alpha = 0.5$).

### 6.5 Impact of non-i.i.d. level

We conducted experiments comparing different degrees of non-i.i.d.-ness, and the results are based on a general setting which is introduced in section 6.1.3. To create non-i.i.d. data with varying degrees, we divided the dataset into 100, 200, and 500 partitions for high, medium, and low degrees of non-i.i.d.-ness, respectively. Then, we randomly selected one partition for the high degree, two partitions for the medium degree, and five partitions for the low degree. In all cases, each client's dataset comprises 500 samples.

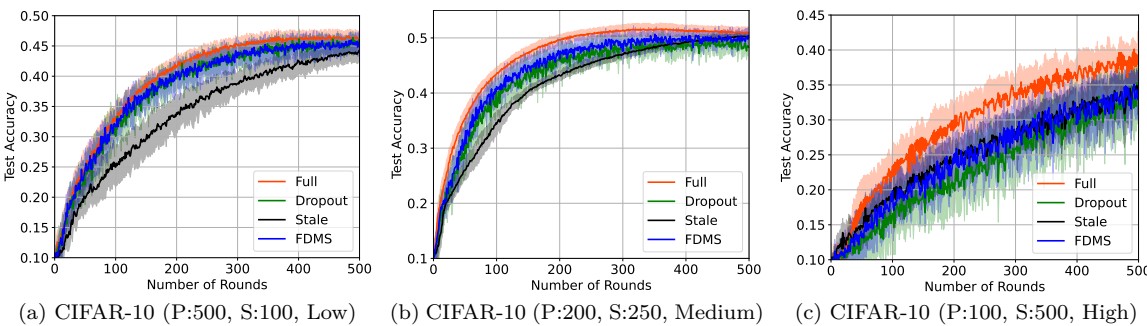

(a) CIFAR-10 (P:500, S:100, Low)    (b) CIFAR-10 (P:200, S:250, Medium)    (c) CIFAR-10 (P:100, S:500, High)

Figure 5: Performance comparison on the CIFAR-10 general setting ($\alpha = 0.5$) with various non-i.i.d. level

The results are presented in Fig. 5. We observed that as the level of non-i.i.d.-ness increased, our method **FL-FDMS** became less effective compared to **Full**, likely due to the increased difficulty in finding friends to perform the model substitution. Nonetheless, **FL-FDMS** still outperformed both the **Dropout** and **Stale** benchmarks, demonstrating that model substitution can enhance convergence compared to doing nothing or using an outdated model.

## 7   Conclusion

This paper investigated the impact of client dropout on the convergence of FL. Our analysis treats client dropout as a special case of local update substitution and characterizes the convergence bound in terms of the total substitution error. This inspired us to develop FL-FDMS, which discovers friend clients on-the-fly and uses friends' updates to reduce substitution errors, thereby mitigating the negative impact of client dropout. Extensive experiment results show that discovering the client's "friendship" is possible and it can be a useful resort for addressing client dropout problems.

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

## A  Proofs

### A.1  Proof of Lemma 2

Consider a dropout client $k$ and a non-friend client $i \notin \mathcal{B}_k$. We analyze the probability that it is selected by the algorithm. According to our selection rule, $i$ is selected only if it has the highest similarity score with client $k$ so far. Hence, $R_t^{i,k}$ must be greater than $R_t^{j,k}$ for at least one $j \in \mathcal{B}_k \cap \mathcal{S}_t$. Thus, the following inequality holds

$$\Pr\{\phi_t(k) = i\} \leq \Pr\{R_t^{i,k} \geq R_t^{j,k}, \text{for some } j \in \mathcal{B}_k \cap \mathcal{S}_t\} \tag{18}$$

$$\leq \Pr\{R_t^{i,k} \geq \mu^{i,k} + \frac{\delta}{2}\} + \Pr\{R_t^{j,k} \leq \mu^{j,k} - \frac{\delta}{2}\} \tag{19}$$

$$\leq \exp\left(\frac{-N_t^{i,k}\delta^2}{2}\right) + \exp\left(\frac{-N_t^{j,k}\delta^2}{2}\right) \tag{20}$$

$$\leq 2\exp\left(\frac{-\beta\delta_k^2 t}{2}\right) \tag{21}$$

where $\delta = \mu^{j,k} - \mu^{i,k} \geq \delta_k$. Because the number of non-friend clients of a client $k$ is at most $K$, the probability of selecting a non-friend client is thus upper-bounded by $2K\exp\left(\frac{-\beta\delta_k^2 t}{2}\right)$. Taking into account $\delta_{\min} = \min_k \delta_k$ completes the proof.

### A.2  Proof of Theorem 1

In this section, we give the proofs in detail. Due to the smoothness in Assumption 1, taking the expectation of $f(w_{t+1})$ over the randomness in round $t$, we have

$$\mathbb{E}_t[f(w_{t+1})] \tag{22}$$

$$\leq f(w_t) + \langle \nabla f(w_t), \mathbb{E}_t[w_{t+1} - w_t]\rangle + \frac{L}{2}\mathbb{E}_t[\|w_{t+1} - w_t\|^2] \tag{23}$$

$$= f(w_t) + \langle \nabla f(w_t), \mathbb{E}_t[\eta\eta_L\Delta_t + \eta\eta_L E\nabla f(w_t) - \eta\eta_L E\nabla f(w_t)]\rangle + \frac{L}{2}\eta^2\eta_L^2\mathbb{E}_t[\|\Delta_t\|^2] \tag{24}$$

$$= f(w_t) - \eta\eta_L E\|\nabla f(w_t)\|^2 + \eta\underbrace{\langle \nabla f(w_t), \mathbb{E}[\eta_L\Delta_t + \eta_L E\nabla f(w_t)]\rangle}_{A_1} + \frac{L}{2}\eta^2\eta_L^2\underbrace{\mathbb{E}_t[\|\Delta_t\|^2]}_{A_2} \tag{25}$$

Note that the term $A_1$ can be bounded as follows:

$$A_1 = \langle \nabla f(w_t), \mathbb{E}_t[\eta_L \Delta_t + \eta_L E \nabla f(w_t)] \rangle \tag{26}$$

$$= \langle \nabla f(w_t), \mathbb{E}_t[\eta_L \bar{\Delta}_t + \eta_L e_t + \eta_L E \nabla f(w_t)] \rangle \tag{27}$$

$$= \left\langle \nabla f(w_t), \mathbb{E}_t \left[ -\frac{1}{K} \sum_{k=1}^{K} \sum_{\tau=0}^{E-1} \eta_L \nabla F^k(w_{t,\tau}^k) + \eta_L e_t + \eta_L E \frac{1}{K} \sum_{k=1}^{K} \nabla F^k(w_t) \right] \right\rangle \tag{28}$$

$$= \left\langle \sqrt{\eta_L E} \nabla f(w_t), -\frac{\sqrt{\eta_L}}{K\sqrt{E}} \mathbb{E}_t \left[ \sum_{k=1}^{K} \sum_{\tau=0}^{E-1} (\nabla F^k(w_{t,\tau}^k) - \nabla F^k(w_t)) - K e_t \right] \right\rangle \tag{29}$$

$$\overset{(a_1)}{=} \frac{\eta_L E}{2} \|\nabla f(w_t)\|^2 + \frac{\eta_L}{2EK^2} \mathbb{E}_t \left\| \sum_{k=1}^{K} \sum_{\tau=0}^{E-1} (\nabla F^k(w_{t,\tau}^k) - \nabla F^k(w_t)) - K e_t \right\|^2$$

$$- \frac{\eta_L}{2EK^2} \mathbb{E}_t \left\| \sum_{k=1}^{K} \sum_{\tau=0}^{E-1} \nabla F^k(w_{t,\tau}^k) - K e_t \right\|^2 \tag{30}$$

$$\overset{(a_2)}{\leq} \frac{\eta_L E}{2} \|\nabla f(w_t)\|^2 + \frac{\eta_L}{EK^2} \mathbb{E}_t \left\| \sum_{k=1}^{K} \sum_{\tau=0}^{E-1} (\nabla F^k(w_{t,\tau}^k) - \nabla F^k(w_t)) \right\|^2$$

$$- \frac{\eta_L}{2EK^2} \mathbb{E}_t \left\| \sum_{k=1}^{K} \sum_{\tau=0}^{E-1} \nabla F^k(w_{t,\tau}^k) - K e_t \right\|^2 + \frac{\eta_L \mathbb{E}_t \|e_t\|^2}{E} \tag{31}$$

$$\overset{(a_3)}{\leq} \frac{\eta_L E}{2} \|\nabla f(w_t)\|^2 + \frac{\eta_L}{K} \sum_{k=1}^{K} \sum_{\tau=0}^{E-1} \mathbb{E}_t \left\| \nabla F^k(w_{t,\tau}^k) - \nabla F^k(w_t) \right\|^2$$

$$- \frac{\eta_L}{2EK^2} \mathbb{E}_t \left\| \sum_{k=1}^{K} \sum_{\tau=0}^{E-1} \nabla F^k(w_{t,\tau}^k) - K e_t \right\|^2 + \frac{\eta_L \mathbb{E}_t \|e_t\|^2}{E} \tag{32}$$

$$\overset{(a_4)}{\leq} \frac{\eta_L E}{2} \|\nabla f(w_t)\|^2 + \frac{\eta_L L^2}{K} \sum_{k=1}^{K} \sum_{\tau=0}^{E-1} \mathbb{E}_t \left\| w_{t,\tau}^k - w_t \right\|^2$$

$$- \frac{\eta_L}{2EK^2} \mathbb{E}_t \left\| \sum_{k=1}^{K} \sum_{\tau=0}^{E-1} \nabla F^k(w_{t,\tau}^k) - K e_t \right\|^2 + \frac{\eta_L \mathbb{E}_t \|e_t\|^2}{E} \tag{33}$$

$$\overset{(a_5)}{\leq} \eta_L E(\frac{1}{2} + 30\eta_L^2 E^2 L^2) \|\nabla f(w_t)\|^2 + 5\eta_L^3 E^2 L^2 (\rho_L^2 + 6E\rho_G^2)$$

$$- \frac{\eta_L}{2EK^2} \mathbb{E}_t \left\| \sum_{k=1}^{K} \sum_{\tau=0}^{E-1} \nabla F^k(w_{t,\tau}^k) - K e_t \right\|^2 + \frac{\eta_L \mathbb{E}_t \|e_t\|^2}{E} \tag{34}$$

where $(a_1)$ follows from that $\langle \mathbf{x}, \mathbf{y} \rangle = \frac{1}{2}[\|\mathbf{x}\|^2 + \|\mathbf{y}\|^2 - \|\mathbf{x} - \mathbf{y}\|^2]$, $(a_2)$ is due to that $\mathbb{E}\|x_1 + x_2\|^2 \leq 2\mathbb{E}[\|x_1\|^2 + \|x_2\|^2]$, $(a_3)$ is due to that $\mathbb{E}\|x_1 + ... + x_n\|^2 \leq n\mathbb{E}[\|x_1\|^2 + ... \|x_n\|^2]$, $(a_4)$ is due to Assumption 1 and $(a_5)$ follows from Lemma 1.

The term $A_2$ can be bounded as

$$A_2 = \mathbb{E}_t[\|\Delta_t\|^2] = \mathbb{E}_t[\|\bar{\Delta}_t + e_t\|^2] \tag{35}$$

$$\overset{(a_6)}{\leq} 2\mathbb{E}_t\|\bar{\Delta}_t\|^2 + 2\mathbb{E}_t\|e_t\|^2$$

$$\leq \frac{2}{K^2}\mathbb{E}_t\left[\left\|\sum_{k=1}^{K}\sum_{\tau=0}^{E-1} g_{t,\tau}^k\right\|^2\right] + 2\mathbb{E}_t\|e_t\|^2 \tag{36}$$

$$\overset{(a_7)}{\leq} \frac{2}{K^2}\mathbb{E}_t\left[\left\|\sum_{k=1}^{K}\sum_{\tau=0}^{E-1}(g_{t,\tau}^k - \nabla F^k(w_{t,\tau}^k))\right\|^2\right] + \frac{2}{K^2}\mathbb{E}_t\left[\left\|\sum_{k=1}^{K}\sum_{\tau=0}^{E-1}\nabla F^k(w_{t,\tau}^k)\right\|^2\right] + 2\mathbb{E}_t\|e_t\|^2 \tag{37}$$

$$\overset{(a_8)}{\leq} \frac{2E}{K}\rho_L^2 + \frac{4}{K^2}\mathbb{E}_t\left[\left\|\sum_{k=1}^{K}\sum_{\tau=0}^{E-1}\nabla F^k(w_{t,\tau}^k) - Ke_t\right\|^2\right] + \frac{4}{K^2}\mathbb{E}_t\|Ke_t\|^2 + 2\mathbb{E}_t\|e_t\|^2$$

$$= \frac{2E}{K}\rho_L^2 + \frac{4}{K^2}\mathbb{E}_t\left[\left\|\sum_{k=1}^{K}\sum_{\tau=0}^{E-1}\nabla F^k(w_{t,\tau}^k) - Ke_t\right\|^2\right] + 6\mathbb{E}_t\|e_t\|^2 \tag{38}$$

where both $(a_6)$ is due to that $\mathbb{E}\|x_1 + x_2\|^2 \leq 2\mathbb{E}[\|x_1\|^2 + \|x_2\|^2]$, $(a_7)$ follows the fact that $\mathbb{E}[\|\mathbf{x}\|^2] = \mathbb{E}[\|\mathbf{x} - \mathbb{E}\mathbf{x}\|^2] + \|\mathbb{E}\mathbf{x}\|^2$, and $(a_8)$ is due to Assumption 3. Substituting the inequalities of $A_1$ and $A_2$ into the original inequality, we have:

$$\mathbb{E}_t[f(w_{t+1})] \tag{39}$$

$$\leq f(w_t) - \eta\eta_L E\|\nabla f(w_t)\|^2 + \eta\underbrace{\langle\nabla f(w_t), \mathbb{E}[\eta_L\Delta_t + \eta_L E\nabla f(w_t)]\rangle}_{A_1} + \frac{L}{2}\eta^2\eta_L^2\underbrace{\mathbb{E}_t[\|\Delta_t\|^2]}_{A_2} \tag{40}$$

$$\leq f(w_t) - \eta\eta_L E\|\nabla f(w_t)\|^2$$
$$+ \eta\eta_L E(\frac{1}{2} + 30\eta_L^2 E^2 L^2)\|\nabla f(w_t)\|^2 + 5\eta\eta_L^3 E^2 L^2(\rho_L^2 + 6E\rho_G^2)$$
$$- \frac{\eta\eta_L}{2EK^2}\mathbb{E}_t\left\|\sum_{k=1}^{K}\sum_{\tau=0}^{E-1}\nabla F^k(w_{t,\tau}^k) - Ke_t\right\|^2 + \frac{\eta\eta_L\mathbb{E}_t\|e_t\|^2}{E}$$
$$+ \frac{EL\eta^2\eta_L^2}{K}\rho_L^2 + \frac{2L\eta^2\eta_L^2}{K^2}\mathbb{E}_t\left[\left\|\sum_{k=1}^{K}\sum_{\tau=0}^{E-1}\nabla F^k(w_{t,\tau}^k) - Ke_t\right\|^2\right] + 3\eta^2\eta_L^2 L\mathbb{E}_t\|e_t\|^2 \tag{41}$$

$$= f(w_t) - \eta\eta_L E(\frac{1}{2} - 30\eta_L^2 E^2 L^2)\|\nabla f(w_t)\|^2$$
$$+ 5\eta\eta_L^3 E^2 L^2(\rho_L^2 + 6E\rho_G^2) + \frac{EL\eta^2\eta_L^2}{K}\rho_L^2 + \left(\frac{\eta\eta_L}{E} + 3\eta^2\eta_L^2 L\right)\mathbb{E}_t\|e_t\|^2$$
$$- \left(\frac{\eta\eta_L}{2EK^2} - \frac{2L\eta^2\eta_L^2}{K^2}\right)\mathbb{E}_t\left\|\sum_{k=1}^{K}\sum_{\tau=0}^{E-1}\nabla F^k(w_{t,\tau}^k) - Ke_t\right\|^2 \tag{42}$$

$$\overset{(a_9)}{\leq} f(w_t) - c\eta\eta_L E\|\nabla f(w_t)\|^2 + 5\eta\eta_L^3 E^2 L^2(\rho_L^2 + 6E\rho_G^2) + \frac{EL\eta^2\eta_L^2}{K}\rho_L^2 + \left(\frac{\eta\eta_L}{E} + 3\eta^2\eta_L^2 L\right)\mathbb{E}_t\|e_t\|^2 \tag{43}$$

where $(a_9)$ follows from $\left(\frac{\eta\eta_L}{2EK^2} - \frac{2L\eta^2\eta_L^2}{K^2}\right) < 0$ if $\eta\eta_L \leq \frac{1}{4EL}$, and that there exits a constant $c > 0$ satisfying $(\frac{1}{2} - 30\eta_L^2 E^2 L^2) > c > 0$ if $\eta_L < \frac{1}{\sqrt{60}EL}$.

Rearranging and summing from $t = 0, ..., T-1$, we have:

$$\sum_{t=0}^{T-1} c\eta\eta_L E \mathbb{E}\|\nabla f(w_t)\|^2 \tag{44}$$

$$\leq f(w_0) - f(w_T) + TE\eta\eta_L \left[5\eta_L^2 EL^2(\rho_L^2 + 6E\rho_G^2) + \frac{\eta\eta_L L}{K}\rho_L^2\right] + \left(\frac{\eta\eta_L}{E} + 3\eta^2\eta_L^2 L\right)\sum_{t=0}^{T=1} \mathbb{E}_t\|e_t\|^2 \tag{45}$$

which implies,

$$\min_{t=0,...,T-1} \mathbb{E}\|\nabla f(w_t)\|^2 \leq \frac{f_0 - f_*}{c\eta\eta_L ET} + \Phi + \Psi(e_0, ..., e_{T-1}) \tag{46}$$

where

$$\Phi = \frac{1}{c}\left[5\eta_L^2 EL^2(\rho_L^2 + 6E\rho_G^2) + \frac{\eta\eta_L L}{K}\rho_L^2\right] \tag{47}$$

$$\Psi(e_0, ..., e_{T-1}) = \frac{1 + 3\eta\eta_L LE}{cE^2 T}\sum_{t=0}^{T-1} \mathbb{E}_t\|e_t\|^2 \tag{48}$$

This completes the proof.

### A.3 Proof of Theorem 2

A sufficient condition for the bound to hold is that after $T$ FL rounds, no friend of client $k$ was eliminated from $\mathcal{C}_k^t$ by running our algorithm. Thus, we are interested in bounding the probability that any particular friend client $i$ is eliminated in a particular round $t$ before $T$.

$$\Pr(i \text{ is eliminated in round } t) \tag{49}$$

$$\leq \Pr(R_t^{k,j} - R_t^{k,i} \geq \Theta_t, \text{ for some } j \neq i) \tag{50}$$

$$\leq \sum_{j\neq i} \Pr(R_t^{k,j} - R_t^{k,i} \geq \Theta_t) \tag{51}$$

$$\leq K\left(\Pr(R^{k,i} \leq \mu^{k,i} - \frac{\Theta_t - \delta_f}{2}) + \Pr(R^{k,j^*} \geq \mu^{k,j^*} + \frac{\Theta_t - \delta_f}{2})\right) \tag{52}$$

$$\leq 2K\exp\left(\frac{-\beta(\Theta_t - \delta_f)^2 t}{2}\right) = q \tag{53}$$

where $j^*$ is the best friend of client $k$. The last equality holds by letting

$$\Theta_t = \sqrt{\frac{2\ln(2K) - 2\ln q}{\beta t}} + \delta_f \tag{54}$$

Next, the probability that a friend client $i$ is eliminated in any round up to round $T$ is bounded as follows

$$\Pr(i \text{ is eliminated up to round } T) \leq \sum_{t \leq T-1} \Pr(i \text{ is eliminated in round } t) \leq Tq \tag{55}$$

Thus,

$$\Pr(\text{any friend of client } k \text{ is eliminated up to round } T) \leq |\mathcal{B}_k|Tq \tag{56}$$

Furthermore,

$$\Pr(\text{any friend of any client is eliminated up to round } T) \leq K|\mathcal{B}_k|Tq \tag{57}$$

Therefore, by letting $p = K B_{\max} T q$ and

$$\Theta_t = \sqrt{\frac{2 \ln(2K^2 T B_{\max}) - 2 \ln p}{\beta t}} + \delta_f \tag{58}$$

we ensure that the probability that no friend of any client was eliminated from the corresponding candidate set by $T$ is at least $1 - p$. This concludes the proof.

## A.4 Bounds on $\mathbb{E}\|e_t\|^2$

The error bound with client dropout:

$$\mathbb{E}[\|e_t\|^2] = \mathbb{E}\left[ \left\| \frac{1}{K} \sum_{k \in \mathcal{K} \backslash \mathcal{S}_t} (\tilde{\Delta}_t^k - \Delta_t^k) \right\|^2 \right] = \mathbb{E}\left[ \left\| \frac{1}{K} \sum_{k \in \mathcal{K} \backslash \mathcal{S}_t} \frac{1}{S_t} \sum_{k' \in \mathcal{S}_t} (\Delta_t^{k'} - \Delta_t^k) \right\|^2 \right] \tag{59}$$

$$\leq \frac{(K - S_t)^2}{K^2} \sigma_P^2 \leq \alpha^2 \sigma_P^2 \tag{60}$$

The error bound with friend model substitution (full information) :

$$\mathbb{E}[\|e_t\|^2] = \mathbb{E}\left[ \left\| \frac{1}{K} \sum_{k \in \mathcal{K} \backslash \mathcal{S}_t} (\tilde{\Delta}_t^k - \Delta_t^k) \right\|^2 \right] = \mathbb{E}\left[ \left\| \frac{1}{K} \sum_{k \in \mathcal{K} \backslash \mathcal{S}_t} (\Delta_t^{\phi_t(k)} - \Delta_t^k) \right\|^2 \right] \tag{61}$$

$$\leq \frac{(K - S_t)^2}{K^2} \sigma_F^2 \leq \alpha^2 \sigma_F^2 \tag{62}$$

where $\phi_t(k)$ is a friend of $k$ that does not dropout in round $t$.

The error bound with friend model substitution (learning):

$$\mathbb{E}\|e_t\|^2 = \mathbb{E}\left\| \frac{1}{K} \sum_{k \in \mathcal{K} \backslash \mathcal{S}_t} (\tilde{\Delta}_t^k - \Delta_t^k) \right\|^2 \leq \frac{K - S_t}{K^2} \sum_{k \in \mathcal{K} \backslash \mathcal{S}_t} \mathbb{E}\|\tilde{\Delta}_t^k - \Delta_t^k\|^2 \tag{63}$$

$$\leq \frac{K - S_t}{K^2} \sum_{k \in \mathcal{K} \backslash \mathcal{S}_t} \left( 2K \exp\left(\frac{-\beta \delta_k^2 t}{2}\right) \sigma_P^2 + \left(1 - 2K \exp\left(\frac{-\beta \delta_k^2 t}{2}\right)\right) \sigma_F^2 \right) \tag{64}$$

$$\leq \frac{(K - S_t)^2}{K^2} \left( \sigma_F^2 + 2K \exp\left(\frac{-\beta \delta_{\min}^2 t}{2}\right) (\sigma_P^2 - \sigma_F^2) \right) \tag{65}$$

$$\leq \alpha^2 \left( \sigma_F^2 + 2K \exp\left(\frac{-\beta \delta_{\min}^2 t}{2}\right) (\sigma_P^2 - \sigma_F^2) \right) \tag{66}$$

### A.5 Bounds on $\Psi(e_0, ..., e_{T-1})$ with friend model substitution (learning)

$$\Psi(e_0, ..., e_{T-1}) \tag{67}$$

$$= \frac{1 + 3\eta\eta_L LE}{cE^2 T} \sum_{t=0}^{T-1} \mathbb{E}_t[\|e_t\|^2] \tag{68}$$

$$\leq \frac{\alpha^2 \sigma_F^2 (1 + 3\eta\eta_L LE)}{cE^2} + 2K \frac{\alpha^2 (\sigma_P^2 - \sigma_F^2)(1 + 3\eta\eta_L LE)}{cE^2 T} \sum_{t=0}^{T-1} \exp\left(\frac{-\beta\delta_{min}^2 t}{2}\right) \tag{69}$$

$$\leq \frac{\alpha^2 \sigma_F^2 (1 + 3\eta\eta_L LE)}{cE^2} + 2K \frac{\alpha^2 \sigma_P^2 (1 + 3\eta\eta_L LE)}{cE^2 T} \sum_{t=0}^{T-1} \exp\left(\frac{-\beta\delta_{min}^2 t}{2}\right) \tag{70}$$

$$\leq \frac{\alpha^2 \sigma_F^2 (1 + 3\eta\eta_L LE)}{cE^2} + 2K \frac{\alpha^2 \sigma_P^2 (1 + 3\eta\eta_L LE)}{cE^2} \frac{1 - \exp\left(\frac{-\beta\delta_{\min}^2 T}{2}\right)}{T\left(1 - \exp\left(\frac{-\beta\delta_{\min}^2}{2}\right)\right)} \tag{71}$$

$$\leq \Psi^* + 2K\bar{\Psi} \frac{1 - \exp\left(\frac{-\beta\delta_{\min}^2 T}{2}\right)}{T\left(1 - \exp\left(\frac{-\beta\delta_{\min}^2}{2}\right)\right)} \tag{72}$$

## B The Relaxed Friend Presence Case

In this section, we consider a relaxed case without Assumption 5. Suppose that a friend of the dropout client is present in each round with a probability $1-r$, then the probability that our algorithm selects a non-friend client for a dropout client in round $t$ is upper bounded by $(1-r)2K \exp\left(\frac{-\beta\delta_{\min}^2 t}{2}\right) + r$. Then we can get the bound on $\mathbb{E}\|e_t\|^2$:

$$\mathbb{E}\|e_t\|^2 \leq \alpha^2 \left(\sigma_F^2 + 2(1-r)K \exp\left(\frac{-\beta\delta_{\min}^2 t}{2}\right)\sigma_P^2 + r\sigma_P^2\right) \tag{73}$$

Plugging this bound into $\Psi(e_0, ..., e_{T-1})$, we can get the accumulate substitution error as follows:

$$\Psi(e_0, ..., e_{T-1}) \leq \Psi^* + r\bar{\Psi} + 2(1-r)K\bar{\Psi} \frac{1 - \exp\left(\frac{-\beta\delta_{\min}^2 T}{2}\right)}{T\left(1 - \exp\left(\frac{-\beta\delta_{\min}^2}{2}\right)\right)} \tag{74}$$

The convergence bound without Assumption 5 has an additional constant term resulting from friend absence, and the additional constant term cannot be eliminated with time or batch size increase.

**Proof 1** *The error bound with friend model substitution (learning) under the relaxed friend presence case is*

$$\mathbb{E}\|e_t\|^2 = \mathbb{E}\left\|\frac{1}{K} \sum_{k \in \mathcal{K}\backslash\mathcal{S}_t} (\tilde{\Delta}_t^k - \Delta_t^k)\right\|^2 \leq \frac{K - S_t}{K^2} \sum_{k \in \mathcal{K}\backslash\mathcal{S}_t} \mathbb{E}\|\tilde{\Delta}_t^k - \Delta_t^k\|^2 \tag{75}$$

$$\leq \alpha^2 \left((1-r)2K \exp\left(\frac{-\beta\delta_{\min}^2 t}{2}\right) + r\right)\sigma_P^2 + \alpha^2 \left(1 - \left((1-r)2K \exp\left(\frac{-\beta\delta_{\min}^2 t}{2}\right) + r\right)\right)\sigma_F^2 \tag{76}$$

$$\leq \alpha^2 \sigma_F^2 + \alpha^2 \left((1-r)2K \exp\left(\frac{-\beta\delta_{\min}^2 t}{2}\right) + r\right)\sigma_P^2 \tag{77}$$

$$\leq \alpha^2 \left(\sigma_F^2 + 2(1-r)K \exp\left(\frac{-\beta\delta_{\min}^2 t}{2}\right)\sigma_P^2 + r\sigma_P^2\right) \tag{78}$$

And the corresponding accumulated substitution error equals

$$\Psi(e_0, ..., e_{T-1}) \tag{79}$$

$$=\frac{1 + 3\eta\eta_L LE}{cE^2 T} \sum_{t=0}^{T-1} \mathbb{E}_t[\|e_t\|^2] \tag{80}$$

$$\leq \frac{\alpha^2(\sigma_F^2 + r\sigma_P^2)(1 + 3\eta\eta_L LE)}{cE^2} + 2K\frac{\alpha^2(\sigma_P^2 - \sigma_F^2)(1 + 3\eta\eta_L LE)}{cE^2 T} \sum_{t=0}^{T-1} \exp\left(\frac{-\beta\delta_{min}^2 t}{2}\right) \tag{81}$$

$$\leq \frac{\alpha^2(\sigma_F^2 + r\sigma_P^2)(1 + 3\eta\eta_L LE)}{cE^2} + 2K\frac{\alpha^2\sigma_P^2(1 + 3\eta\eta_L LE)}{cE^2 T} \sum_{t=0}^{T-1} \exp\left(\frac{-\beta\delta_{min}^2 t}{2}\right) \tag{82}$$

$$\leq \frac{\alpha^2(\sigma_F^2 + r\sigma_P^2)(1 + 3\eta\eta_L LE)}{cE^2} + 2K\frac{\alpha^2\sigma_P^2(1 + 3\eta\eta_L LE)}{cE^2} \frac{1 - \exp\left(\frac{-\beta\delta_{min}^2 T}{2}\right)}{T\left(1 - \exp\left(\frac{-\beta\delta_{min}^2}{2}\right)\right)} \tag{83}$$

$$\leq \Psi^* + r\bar{\Psi} + 2K\bar{\Psi}\frac{1 - \exp\left(\frac{-\beta\delta_{min}^2 T}{2}\right)}{T\left(1 - \exp\left(\frac{-\beta\delta_{min}^2}{2}\right)\right)} \tag{84}$$

## C  Additional Experiments

### C.1  Impact of number of local iterations $E$

The error bound of FL-FDMS $\mathbb{E}\|e_t\|^2$ in Eq. 16 is influenced by the number of local iterations $E$ and the number of clients $K$. Next, we perform additional experiments to explore their impacts. We present more results on the performance comparison in the MNIST clustered setting and the CIFAR-10 clustered setting with different $E$. We fix $\alpha = 0.5$ and $K = 20$ for all the following experiments. To investigate the impact of $E$, we consider two values $E = 1$ and $E = 5$.

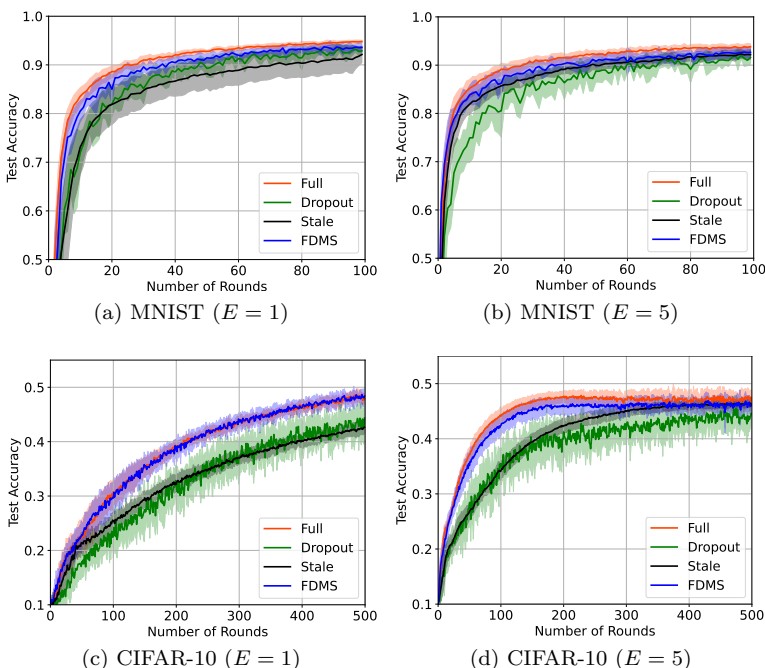

(a) MNIST ($E = 1$)  (b) MNIST ($E = 5$)

(c) CIFAR-10 ($E = 1$)  (d) CIFAR-10 ($E = 5$)

Figure 6: Performance comparison with $\alpha = 0.5$ and $K = 20$

In Fig. 6, we find that the **FL-FDMS** still shows superior performance in terms of test accuracy and convergence speed. However, **Dropout** and **Stale** show different trends for different $E$. For a larger $E$, using staled models tends to help the dropout situation better.

### C.2  Impact of number of clients $K$

To investigate the impact of $K$, we fix $E = 2$ and increase the number of clients to $K = 40$. To keep the same total amount of data in the system, we adjust just the number of data samples on each client. For MNIST, each client now has 100 samples. For CIFAR-10, each client has 500 samples.

By comparing Fig. 7 and the corresponding parts in Fig. 1, we find that as $K$ increases, the **FL-FDMS** outperforms **Dropout** and **Stale** even more. This is because as $K$ increases, more clients dropout. If the model updates from dropout clients are not compensated, the global model can gradually deviate from the optimal value and eventually degrade the learning performance and affect the system's stability. The additional experiments further verify that **FL-FDMS** can handle well the client dropout in FL.

### C.3  Experiments with FedProx

In the Cifar10 clustered setting, we have conducted additional performance comparison experiments using FedProx Li et al. (2020a) with the FedProx parameter value $\mu = 0.2$. Our results, shown in Fig. 8, demonstrate that the proposed **FL-FDMS** algorithm remains effective.

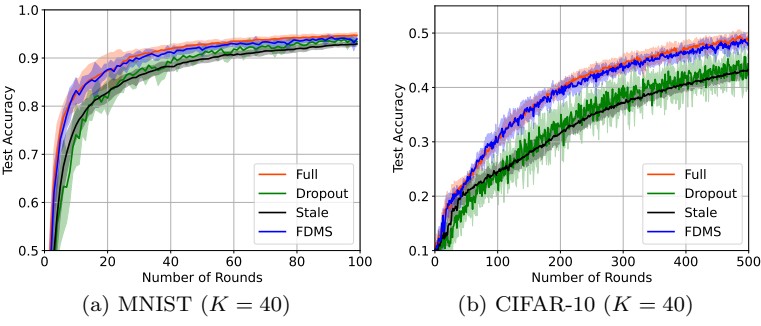

(a) MNIST ($K = 40$)  (b) CIFAR-10 ($K = 40$)

Figure 7: Performance comparison with $\alpha = 0.5$ and $E = 2$

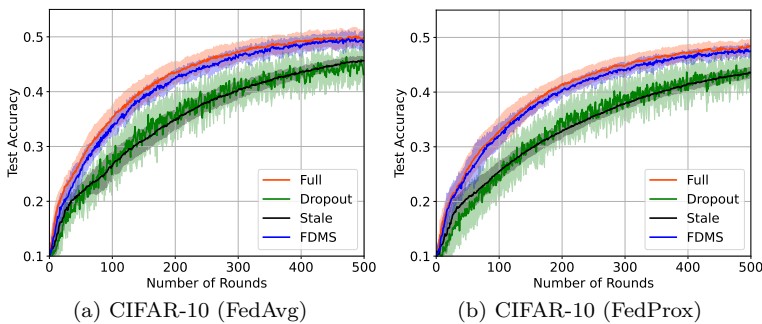

(a) CIFAR-10 (FedAvg)  (b) CIFAR-10 (FedProx)

Figure 8: Performance comparison on the CIFAR-10 clustered setting ($\alpha = 0.5$) with different FL algorithms

## C.4  Experiments on the FMNIST datasets

We present additional performance comparison results in the FMNIST clustered setting, and the results in Fig. 9 are consistent with the conclusions we have drawn from prior other datasets.

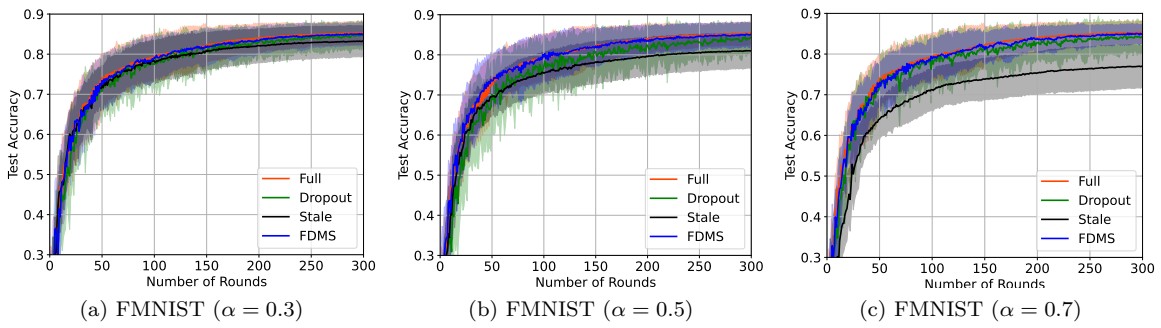

(a) FMNIST ($\alpha = 0.3$)  (b) FMNIST ($\alpha = 0.5$)  (c) FMNIST ($\alpha = 0.7$)

Figure 9: Performance comparison on the FMNIST clustered setting with various $\alpha$

