# OpenReview forum: "Combating Client Dropout in Federated Learning via Friend Model Substitution"
_TMLR — Withdrawn by Authors_

### Review · Reviewer_rA24 · 2023-06-12

**Summary Of Contributions:**

The paper studies federated learning with client dropout for generic update substitution and with a specific update substitution algorithm. The main contributions of the paper are:

1. An analysis of the convergence error of FL with arbitrary update substitution under standard smoothness assumptions (no convexity is assumed). This is instantiated for special cases.

1. An algorithm that discovers "friends" (i.e., clients with similar updates in L2/cosine distance, assuming they exist). The assumptions made are: (i) every dropout client has a "friend" that participates, and (ii) all pairs of clients co-participate frequently enough. Specifically, out of $t$ rounds, every pair $(i, j)$ of clients simultaneously appears in $\Omega(t)$ rounds. Under these assumptions, it is shown that the error from substitution asymptotically equals the one where the friendship structure is known.

1. A computation-efficient version of the above algorithm where clients that are not friends are eliminated.

1. Experiments demonstrate that the proposed method outperforms baselines and nearly matches the case of a known friendship structure.

**Audience:**

Yes

**Broader Impact Concerns:**

The paper needs to address if and how the proposed method exacerbates disparities in client performance. Some additional experimental work is necessary to investigate this aspect (see above).

**Claims And Evidence:**

No

**Requested Changes:**

### Required Changes

I view these changes as necessary for acceptance.

* **Rigor/correctness**: Make the math/proofs rigorous and fix bugs, if any. See the previous section for details

* **Modify/explain strong assumptions**: The authors must weaken the assumptions and/or provide an elaboration of the assumptions made and the conditions under which they hold. See the previous section for details.

* **Experiments**:
	- I cannot find some important details. How are the dropouts simulated? What is the similarity metrics to define friendship?
	- It is important to visualize per-client metrics (e.g. a histogram of all client errors or a scalar summary such as the worst-client error). It is important to make sure that the proposed methods do not exacerbate performance disparities among clients.
	- How is $\Theta_t$ chosen in Section 6.4? See the previous section again for my confusion.



### Recommended changes

The following changes would greatly improve the paper in my view. I encourage the author to make an effort to address them.

* **Discussion of exact setting**:
	- Does the paper target cross-device FL or cross-silo FL? How does the exact setting in the paper map back to previously studied settings? It is ok if the setting of the paper is slightly different, but it would be good to provide the readers with the context.
	- It would greatly help to be precise about the information exchange and threat model. For example, it appears to me that the paper considers a setting where the clients trust the server fully and send their updates in the open. In contrast, the so-called "honest-but-curious" model is more realistic in practice (see Table 7 or Kairouz et al. 2021) but the proposed methods appear to be incompatible with it.

* **Fixing the citations**: correct the typos in names (e.g. konevcny), appropriate use of \citep instead of the \citet, remove redundant/repeated references, cite the published versions of each paper and not the arxiv report, etc.

* **Experimental reporting**: It would help to report simple summaries of experimental results rather than training curves. It is good to have the training curves in the appendix, but it might help readability to summarize each training curve with a number (e.g. last iterate or average over last 5 iterates). It would also help to see these numbers in a table or as a plot with other parameters varying (e.g. $\alpha$ or computational complexity).

* Why is the friendship structure more obvious for CIFAR-10 than MNIST? Does it have to do with the complexity of the model or the number of clients? It would be good to see some more ablations on this subject.

* It would also be good to see some experiments with a natural data split e.g. EMNIST.



### References

Kairouz et al. (2021). Advances and Open Problems in Federated Learning.

**Strengths And Weaknesses:**

### Strengths

1. The paper presents a novel take on client dropout. It is well-written and the arguments are easy to follow.

1. The convergence proofs appear to be correct. I did not check them line by line or check for constants but the arguments make sense.

1. The experiments are well-designed and well-executed. They complement the theoretical arguments nicely.

### Weaknesses

There are critical weaknesses surrounding the mathematical rigor and correctness of proofs.

**Proof step unclear**:
Could you explain how you get Eq. (19) in the proof of Lemma 2? I am not convinced this inequality is correct. For reference, it says
$$
 P(R^{i, k}_t \ge R^{j, k}_t) \le P(R^{i, k}_t \ge \mu^{i, k} + \delta/2) + P(R^{j, k}_t \le \mu^{j, k} - \delta / 2).
$$
The same holds also for Eq. (52) in the proof of Theorem 2.

**Questions on Lemma 2 & Theorem 2 due to lack of rigor**: The conditional expectations are rather sloppy. Sometimes, the conditioning is missing and in several instances, it is not clear what are the expectations defined with respect to. This makes me question the correctness of Lemma 2 and all the conclusions drawn from it. Here are the issues in more detail.

- Firstly, $\mathbb{E}_t$ is not defined. I assume this is the expectation conditioned on all the information known at time $t$. If that is the case, the right side of the bound of Theorem 1 should have an unconditional expectation (a conditional expectation $\mathbb{E}_t[\cdot]$  with this definition is a random variable depending on $w_t$).

- Similarly, what does $\mathbb{E}_t[r^{i, j}_t]$ refer to? Again, I'm assuming that it is an expectation conditioned on all the information up to time $t$. The Hoeffding bound in the proof of Lemma 2 (Eq. 20) assumes that the random variables $r^{i, j}_t$ for different $t$ are independent and identically distributed. However, they are dependent due to the fact that they depend on the models $w_t$ (for different $t$). There is no reason to expect that they are identically distributed either (the expectation is independent of $t$, implicitly assuming that this is the case). If this is the definition of $\mathbb{E}_t$, then the proof is wrong.

- An alternative definition of $\mathbb{E}_t$ above is possible. If the Hoeffding bound above is invoked on the finite collection $r^{i, j}_t$, then $\mu^{i, j}$ is a finite average $R^{i, j}_T$. Then, there is an implicit assumption that all of these are conditioned on the iterates $w_1, \ldots, w_T$, so $\mu^{i, j}$ is a function of the iterates. The assumptions on $\delta$ are now assumed to hold almost surely on a random sequence $w_1, \ldots, w_T$. This is fine, but it must be clearly stated (right now, it is a hidden/implicit assumption).

- In the latter case, $\delta_f$ and $\Theta_t$ in Theorem 2 are not known in advance. What value of $\Theta_T$ is the efficient version run with?


**Strong and unverifiable assumptions**:

- Assumption 4 is on the result of the local updates. This is an unverifiable assumption as it depends on the current iterate $w$ and the dynamics of gradient descent. It would be much more interpretable to have this assumption in terms of the properties of the local objectives directly (e.g. gradient of the local objective). If not in full generality, it would be good to instantiate this for very simple settings such as linear or logistic regression.

- Similarly, Assumption 6 is also very strong and it might be hard to satisfy that for $t$ small. To get an intuition on how strong this condition is, it would be helpful to see the conditions under which it holds with a high probability for purely random dropouts. It may be more meaningful to require this assumption for $t \ge t_0$ for some threshold $t_0$. Similarly, it may also make sense to require a slower rate of growth, e.g. $t^a$ for $0< a < 1$.

---

### Review · Reviewer_3wHK · 2023-06-17

**Summary Of Contributions:**

The paper presents FL-FDMS, a Federated Learning (FL) framework designed to address the critical problem of client dropouts (or partial participation) during training.
This novel method aims to tackle this issue by replacing the model of the dropout clients with the model of their "friends," clients with a similar data distribution. The model employs the Friend Discovery Mechanism (FDM), where friends are identified using the Pairwise Cosine Similarity (PCS) and are dynamically updated throughout the iterations procedure. The authors also introduced a practical mechanism that computes a group of "friends".
The paper posits that by substituting dropout clients with their friends, FL-FDMS can achieve a faster convergence while maintaining a performance close to an ideal full-participation scenario.

For smooth non-convex functions, authors presented
$\mathcal{O}\left(\frac{1}{E \sqrt{T}}\right)+\mathcal{O}\left(\frac{E^2}{T}\right)+\mathcal{O}\left(\frac{1}{K \sqrt{T}}\right)+\mathcal{O}\left(\frac{\alpha^2 \sigma_P^2}{\sqrt{T}}\right)+\mathcal{O}\left(\frac{\alpha^2 \sigma_P^2}{E}\right)$
convergence rate,
where $E$ is the number of local steps,
$K$ - number of clients,
$T$- number of communications,
$\alpha$ - participation ratio.

Experiments are conducted using Python3 and Pytorch library on two standard public datasets, MNIST and CIFAR-10, in both clustered and general settings. The clustered settings were used to evaluate the friend discovery performance of FL-FDMS in a controlled environment, and the general setting was to test the method's applicability on a commonly used non-IID FL dataset.
The authors chose CNN models for both datasets, with various configurations for each. Three versions of FedAvg, Full Participation (Full), Client Dropout (Dropout), and Staled Substitute (Stale), were chosen as benchmarks for comparison with FL-FDMS. The method consistently outperformed Dropout and Stale, achieving performance close to Full. FL-FDMS showed a particularly significant performance improvement on datasets like CIFAR-10 and MNIST.
Additionally, FL-FDMS showed successful friend discovery. The pairwise similarity scores in the final learning round demonstrated that intra-cluster client pairs had larger similarity scores, indicating successful clustering or friendship discovery. Furthermore, the authors presented the effectiveness of the complexity reduction mechanism by demonstrating a significant reduction in computational complexity without considerable impact on learning performance.
However, the experiments revealed a limitation: FL-FDMS's performance decreased with an increase in non-IID-ness. Even though it still outperformed Dropout and Stale benchmarks, it fell short of the Full Participation method. This suggests that the algorithm is most effective when the "friendship" relationship among the clients is stronger.

**Audience:**

Yes

**Broader Impact Concerns:**

This work is primarily theoretical in nature.
Hence, there is no identifiable potential for negative societal impact arising from this work.


**Claims And Evidence:**

Yes

**Requested Changes:**

**Suggestions**
1) I would suggest that the authors address the following typographical error found within the text:
 On page 3, the phrase: "We consider a typical FL algorithm konevcny et al. (2016)":
The surname "konevcny" begins with a lowercase letter and appears to be misspelled.
2) Regarding weakness 1) previously mentioned, I would recommend adding a discussion where the provided theoretical rate is contrasted with other relevant baselines, such as without replacement uniform sampling (commonly used by theoreticians [1,2]) or other strategies [3]. For instance, Chen. et al [3] proposed an (in a certain sense) optimal client sampling, the performance of which provably lies between uniform sampling and full participation.
3) As the primary motivation for the proposed client sampling strategy is derived from the experimental performance, I would suggest comparing it with an expanded set of baselines on a broader array of tasks (for example, language tasks). It would be worthwhile to include at least paper [3] in the comparison.
4) With respect to experiments, it is also standard practice to compare communication-efficient methods in terms of the number of bits transmitted from the workers to the server. This measure is somewhat proportional to real-time computation or real-time work (see [1,2,3,4,5]).
5) I would suggest adding some plots that compare experiments based on this measure.

[1] Chen, et al. "Optimal client sampling for federated learning." (arXiv:2010.13723), 2020.

[2] Zhao, et al. "Faster Rates for Compressed Federated Learning with Client-Variance Reduction." (arXiv:2112.13097), 2021.

[3] Tyurin, et al. "A Computation and Communication Efficient Method for Distributed Nonconvex Problems in the Partial Participation Setting." (arXiv:2205.15580), 2022.

[4] Fatkhullin, et al. "EF21 with bells & whistles: practical algorithmic extensions of modern error feedback." (arXiv:2110.03294), 2021.

[5] Richtárik, et al. "EF21: A new, simpler, theoretically better, and practically faster error feedback." In: Advances in Neural Information Processing Systems, 2021.

**Strengths And Weaknesses:**

**Strengths:**
1) The main paper is clearly written, with its principal claims effectively outlined, making the text easy to read;
2) The literature review contains pertinent papers related to the topic;
3) The authors propose a novel, practically motivated client selection strategy. This is fed into the classic federated averaging algorithm. Also, the authors derived convergence rates in the smooth non-convex regime. In addition, they provide extensive experimental results where the proposed sampling strategy is benefited over existing baselines in some regimes.

**Weaknesses:**
1) While the authors provide a comprehensive list of pertinent references, the theoretical results proposed in this paper lack comparison with existing baselines. It would be intriguing to observe how the provided convergence bounds compare to the theoretical bounds of existing baselines involving client sampling. Currently, only a comparison with the full participation scenario is available.
2) This same issue extends to the experimental setup. Please see "Requested Changes" section, point 2)3).
3) The description of the experimental setup requires some clarification. For instance, it is unclear from the text how "dropout" samples clients.
4) On pages 9 and 10, "the local learning rate $\eta_L = 0.1$ and the global learning rate $\eta = 0.1$" are mentioned. The paper does not explicitly explain why these particular global and local learning rates were selected. In deep learning experiments, fine-tuning typically takes place to identify the appropriate step size for each method. This is critical since different methods may require different step sizes, meaning a step size optimal for one method may not necessarily be optimal for another. Without learning rate fine-tuning, the comparison of different methods (or client sampling schemes in this case) is of limited relevance.
5) The phrase "since typically $\sigma_F^2 \ll \sigma_P^2$" ( where $\sigma_P^2 = \max_{i, j} \sigma_{i, j}^2$ ) prompts some doubts. Can you clarify whether this holds either in theory or practice? Results of this nature are not presented in the paper. It is also unclear whether $\sigma_p$ can be estimated beforehand since it relies on the somewhat restrictive Assumption 4.

*Assumption 4: For any two clients $i$ and $j$, the local model update difference is bounded as follows:
$$
\mathbb{E}\left[\left\|\Delta^i(w)-\Delta^j(w)\right\|^2\right] \leq \sigma_{i, j}^2, \forall w
$$
where the expectation is over the local dataset samples.*

6) Assumption 5, which introduces the constant $\sigma_F$, appears equally restrictive and difficult to verify.

   *Definition 1 (Friendship): Let $\sigma_F^2<\sigma_P^2$ be some constant. We say that clients $i$ and $j$ are friends if $\sigma_{i, j}^2 \leq \sigma_F^2$. Further, denote $\mathcal{B}_k$ as the set of friends of client $k$ and $B_k=\left|\mathcal{B}_k\right|$ as the size of $\mathcal{B}_k$.*

*Assumption 5 (Friend Presence): In any round $t$, for any inactive client $i \in \mathcal{K} \backslash \mathcal{S}_t$, there exists at least one active client $j$ that is client $i$ 's friend.*



7) The same concern holds for the minibatch case, i.e., the assumption $\mathbb{E}\left[\left\|\Delta^i(w)-\Delta^j(w)\right\|^2\right] \leq \sigma_F^2 / B, \forall w$
8) While the sampling is practically motivated, it doesn't seem compatible with standard system-level practices like secure aggregation [6, 7, 8, 9], as it requires access to individual updates, it introduces a privacy concern.

[6] Du, et al. "Secure multi-party computation problems and their applications: a review and open problems." In: Workshop on New Security Paradigms, 2001.

[7] Kairouz, et al. "Advances and open problems in federated learning." (arXiv:1912.04977), 2019.

[8] Bonawitz, et al. "Practical secure aggregation for privacy-preserving machine learning." In: Proceedings of the 2017 ACM SIGSAC Conference on Computer and Communications Security, pp. 1175–1191, 2017.

[9] So, et al. "Turbo-aggregate: Breaking the quadratic aggregation barrier in secure federated learning." In: IEEE Journal on Selected Areas in Information Theory, 2(1):479–489, 2021.

---

### Review · Reviewer_r8KH · 2023-07-04

**Summary Of Contributions:**

This paper considers the problem of client dropout in cross-device Federated Learning, where clients may indicate that they are available at the beginning of a round but then not respond by the end of the round for one of a variety of reasons (e.g., loosing access to a power source or communication connection). The paper proposes an approach where updates from clients that dropped out are replaced with updates from clients with similar data distributions -- so-called "friends" in this work. A simple approach is proposed to calculate friends by tracking update similarities across rounds. Theoretical convergence guarantees are provided, and experiments illustrate the promise of the proposed approach.

**Audience:**

No

**Broader Impact Concerns:**

None noted

**Claims And Evidence:**

No

**Requested Changes:**

The paper claims to address a practical concern in cross-device FL. In order to be convinced that this is indeed the case, I would need much stronger evidence supporting that the assumptions made in the paper are valid in typical cross-device FL settings that have been outlined in previous work. In particular, the paper should explain how pair-wise similarity scores can be calculated in a privacy-preserving manner, it should justify why clients are participating many times throughout training, and it should quantify the additional overhead required at the server and justify why that may be feasible.


**Strengths And Weaknesses:**

## Strengths
1. The problem of dealing with client dropout in cross-device FL considered in this paper is relevant problem to those deploying FL systems.
2. The theoretical results appear to be rigorous and sound.
3. The paper is well-written and easy to follow.

## Weaknesses

1. The primary weakness is that the setup assumed in this work is very disconnected from scenarios of current practical interest in cross-device FL; see See [Bonawitz et al., (2019)](https://arxiv.org/abs/1902.01046) and [Huba et al., (2021)](https://arxiv.org/abs/2111.04877).
    * Namely, in cross-device FL systems today, there are generally many millions of clients participating in training (up to hundreds of millions have been reported). In this regime, each client typically participates at most once (i.e., in at most one round) of training. This would make it impossible to satisfy the Sufficiently Many Common Rounds assumption.
    * Furthermore, for privacy considerations, the server should never receive or process an update from an individual client; rather _secure aggregation_ is used so that the server only observes updates in aggregate. If secure aggregation is being used, the server would not be able to compute any pair-wise similarity score. If secure aggregation is not used, then it is not clear how privacy is preserved any more, and much additional discusion about privacy is required. The remark at the end of Section 5.1 is not sufficient, since it is well-know that if an adversary has access to model updates from an individual client then it can recover that client's training data through a model inversion attack.

2. One approach to handling stragglers which was not mentioned in the related work section is to use asynchronous updating, as in FedAsync and FedBuff/Papaya. The other most commonly implemented approach for synchronous FL methods is to incorporate some sort of timeout or use "over-selection" (see Bonawitz et al., 2019).

3. Storing and maintaining pair-wise similarity scores for all pairs of clients would require memory at the server that is quadratic in the number of clients, and this is infeasible for typical cross-device FL systems based on the number of clients typically involved in training.


As an aside, client dropout is a much more significant challenge for secure aggregation schemes (used to preserve privacy, so no individual clients' update is visible to the server or any other client) than it is a hamper to the overall algorithm convergence.

---

### Note · Authors · 2023-07-21

I have read and agree with the venue's withdrawal policy on behalf of myself and my co-authors.